# Modulation of Autophagy on Cinnamaldehyde Induced THP-1 Cell Activation

**DOI:** 10.3390/ijms26062377

**Published:** 2025-03-07

**Authors:** Yi Qin, Fan Wu, Rui Wang, Jun Wang, Jiaqi Zhang, Yao Pan

**Affiliations:** 1Department of Cosmetics, School of Light Industry Science and Engineering, Beijing Technology and Business University, Beijing 100048, China; 2330402109@st.btbu.edu.cn (Y.Q.); 2230401031@st.btbu.edu.cn (F.W.); wangrui09150127@163.com (R.W.); 2330402112@st.btbu.edu.cn (J.W.); 2230402137@st.btbu.edu.cn (J.Z.); 2Beijing Key Laboratory of Plant Research and Development, Beijing 100048, China

**Keywords:** cinnamaldehyde, autophagy, THP-1 cell, skin sensitization, DC activation

## Abstract

Cinnamaldehyde (CIN), which is a cosmetic fragrance allergen regulated by the European Union, can induce allergic contact dermatitis in consumers, reducing their quality of life. Autophagy may be associated with the dendritic cell (DC) response to chemical sensitizers. We hypothesized that CIN would activate DCs through autophagy during skin sensitization. In this study, Tohoku Hospital Pediatrics-1 cells (THP-1 cells) were used as an in vitro DC model, and we evaluated the expression of cell activation markers, intracellular oxidative stress, and autophagy pathway-related genes in response to CIN in THP-1 cells. CIN exposure activated THP-1 cells, which presented increases in CD54 and CD86 expression and ROS generation. Transcriptomic analysis revealed that the genes that were differentially expressed after CIN stimulation were mostly associated with autophagy. The autophagy markers LC3B, p62, and ATG5 had upregulated mRNA and protein levels after CIN exposure. Furthermore, the effects of the autophagy inhibitor Baf-A1 and the autophagy activator rapamycin were investigated on CIN-treated cells. Pretreatment with Baf-A1 in THP-1 cells impaired autophagic flux and dramatically promoted cell activation and oxidative stress triggered by CIN. Conversely, rapamycin inhibited cell activation and the ROS content in CIN-challenged cells while increasing autophagy levels via a reduction in mTOR expression. These results suggest that the autophagy pathway has a pivotal influence on the regulation of CIN-induced activation in THP-1 cells, which provides new insight into the pathogenesis and precise therapeutic strategies for ACD.

## 1. Introduction

Allergic contact dermatitis (ACD) is an inflammatory skin disease affecting approximately 20% of the general population due to repeated exposure to a minimum of one contact allergen [1]. ACD manifests as delayed-type hypersensitivity characterized by intensely pruritic erythema, edema and vesicles, which strongly affects patients’ quality of life [2]. Previous patch-testing studies have shown that a variety of substances contained in healthcare products can cause ACD, including fragrance mixtures, isothiazolinones, and paraphenylenediamine, with fragrance mixtures being the main cause of sensitization [3,4]. Cinnamaldehyde (CIN), a hydroxyacid fragrance-containing compound, is one of the 26 fragrance allergens regulated by the European Union and must be listed on the labels of cosmetics and household products [5,6,7,8]. The prevalence of sensitization to CIN in routine testing ranges from 1.2% to 1.9%, and that is 0.8% in the general population, which is among the fragrance with the highest reactivity in the 26 allergens [9,10]. Additionally, occupational ACD case reports have shown that bakers frequently exposed to cinnamon have positive patch test results for CIN [11,12]. Given its high sensitization potential and widespread use, CIN was chosen as the focus of this study to investigate the mode of action of ACD.

The relevant mechanism of skin sensitization is summarized in the Organization for Economic Cooperation and Development (OECD) report as the adverse outcome pathway (AOP), which is divided into four key events (KEs), including covalent binding of hapten to proteins (KE1), activation of keratinocytes (KE2), activation of dendritic cells (DCs) (KE3), and proliferation and differentiation of T cells (KE4) [13,14,15,16]. DCs capture hapten–protein conjugates and migrate to skin-draining lymph nodes, where they present haptenized peptides to activate T cells [17]. Contact allergens stimulate DC maturation to increase their antigen-presenting ability by increasing the expression of costimulatory molecules (e.g., CD54 and CD86), as well as the production of the proinflammatory cytokine IL-8 [18,19]. Furthermore, studies have revealed that autophagy might play a role in DCs during skin sensitization. A proteomic-based evaluation of the skin sensitization hazard of glyphosate demonstrated that the test compounds could interfere with autophagy flux, as they induce sequestosome 1 (SQSTM1/p62) protein expression in a DC model [20]. The classical sensitizer 1-fluoro-2,4-dinitrobenzene (DNFB) reportedly regulates the maturation of Tohoku Hospital Pediatrics-1 cells (THP-1 cells), a DC-like cell line, and stabilizes the antioxidant molecule Nrf2 via the induction of two autophagy-related genes, microtubule-associated protein 1 light chain 3B (LC3B) and p62 [21]. However, while these studies reported this phenomenon, the underlying mechanisms connecting autophagy and DC activation in skin sensitization remain unexplored.

Autophagy ensures cellular homeostasis by degrading and recycling damaged components. First, phagophores form, in which autophagy-related protein 5 (ATG5) facilitates LC3 lipidation (LC3-I to LC3-II), and LC3-II integrates into the autophagosome membrane [22]. LC3 has three isoforms, LC3A, LC3B, and LC3C, with LC3B being the primary isoform for monitoring autophagic flux due to its strong membrane binding ability and reliable lipidation [23]. As the phagophore elongates, it captures damaged cellular components. During this process, p62 helps target damaged proteins and organelles to the autophagosome for degradation [24]. The autophagosome fuses with the lysosome to form the autolysosome, where the contents are degraded as well as p62.

Based on the above findings, we speculated that CIN might activate DCs by triggering autophagy-related pathways, thereby resulting in a skin sensitization reaction. We aimed to provide deeper insights into the impact of autophagy on skin sensitizer-induced DC activation. In this study, we investigated the activation of THP-1 cells induced by CIN and the expression of key regulatory molecules of the autophagy process, including LC3B, p62, and ATG5. Additionally, we examined the impacts of pretreatment with an autophagy inhibitor or activator on the CIN-induced activation of THP-1 cells to clarify the autophagy-related pathway mechanism involved in THP-1 cell activation under CIN exposure. THP-1 cells, a human monocytic leukemia cell line, serve as a standard surrogate for DCs because of their similarity with primary DCs in morphology and differentiation [25]. THP-1 cells can be activated by most skin sensitizers and have been employed in OECD-adopted skin sensitization assays, which confirms that this cell line is a useful model for in vitro skin sensitization screening [26,27].

## 2. Results

### 2.1. CIN-Induced THP-1 Cell Activation

CIN significantly reduced THP-1 cell survival in a dose-dependent manner (Figure 1A). As per the rules of the OECD test guideline (TG) 442E, the computed CV75 for CIN was 1‱. THP-1 cell activation was evaluated at this dose by measuring the relative fluorescence intensity (RFI) of CD54 and CD86. At CIN concentrations above 0.69‱, the RFI of CD86 exceeded 150%, whereas at concentrations above 1‱, the RFI of CD54 exceeded 200% (Figure 1B,C). CIN was categorized as positive in the human cell line activation test (h-CLAT) according to the predictive model. For further research, concentrations of 0.69‱ and 1‱ were chosen, as those increased the expression of CD54 and CD86. We examined the intracellular levels of reactive oxygen species (ROS) in THP-1 cells treated with CIN to further investigate the oxidative conditions of these cells. Compared with the control, CIN exposure dramatically increased the generation of ROS in THP-1 cells (Figure 1D). These findings confirmed that the CIN-induced THP-1 cell activation model was successfully established.

### 2.2. CIN Interfered with the Autophagy Pathway in THP-1 Cells

To investigate the effect of CIN on the THP-1 cell pathway, a transcriptomics method was applied. Transcriptome sequencing results revealed 330 differentially expressed genes in 0.69‱ CIN-treated THP-1 cells compared with the control cells, of which 259 genes were upregulated and 71 genes were downregulated. In 1‱ CIN-treated THP-1 cells, a total of 1974 differentially expressed genes were identified when compared to control cells, of which 1071 genes were upregulated and 903 genes were downregulated, as shown in Figure 2A. The results of the KEGG enrichment analysis of the differentially expressed genes are represented by a scatter plot in Figure 2B. KEGG analysis revealed that the differentially expressed genes were involved primarily in signalling pathways related to autophagy, mitochondrial autophagy and ferroptosis. As shown in the heatmap, CIN treatment in THP-1 cells significantly impacted the expression of autophagy-associated genes, especially SQSTM1/p62, LC3B and ATG5 (Figure 2C). These findings suggest that autophagy might be the most disturbed pathway in CIN-exposed THP-1 cells.

To verify the accuracy of the transcriptome data, three representative autophagy-related genes—*LC3B*, *p62*, and *ATG5*—were screened and validated using qRT-PCR and Western blot analysis. The qRT-PCR results indicated that the mRNA expression levels of *LC3B*, *p62*, and *ATG5* were significantly greater in the CIN-treated group than in the control group, as shown in Figure 3A–C. The Western blot results were consistent with the qRT-PCR results, which revealed that the protein expression of LC3B, p62, and ATG5 was increased in the CIN-treated groups (Figure 3D).

### 2.3. The Autophagy Inhibitor Baf-A1 Impaired Autophagic Flux and Increased CIN-Induced Activation in THP-1 Cells

Increased LC3B expression can suggest increased autophagy or impaired autophagic flux. Compared with CIN alone, pretreatment with the late autophagy inhibitor Baf-A1 resulted in increased levels of *LC3B*, *p62*, and *ATG5* (Figure 4A–C). The protein expression levels were consistent with the qRT-PCR results, as shown in Figure 4D.

We further studied the influence of Baf-A1 on CIN-induced activation in THP-1 cells. The results showed that in comparison to non-Baf-A1-treated cells, noncytotoxic doses of Baf-A1 significantly increased the activation level of THP-1 cells, including the levels of the cell surface molecules CD54 and CD86 and ROS levels (Figure 4E–G).

### 2.4. The Autophagy Activator Rapamycin Promoted Autophagy and Decreased CIN-Induced Activation in THP-1 Cells

To determine whether augmenting autophagy could reverse the overaccumulation of p62 following CIN treatment, we used RAPA, which is an autophagy activator and a specific mTOR inhibitor. Compared with CIN exposure alone, pretreatment with RAPA significantly increased the expression of LC3B and ATG5 at both the transcriptional and protein levels while markedly reducing that of p62 (Figure 5A–D). In addition, mTOR protein expression dramatically decreased after RAPA treatment (Figure 5D), suggesting that the application of RAPA successfully promoted cellular autophagy.

The enhancement of autophagy in THP-1 cells abolished the expression of CD54 and CD86 induced by CIN, with the RFI much lower than that in the absence of RAPA (Figure 5E,F). Furthermore, CIN-mediated ROS generation was decreased by pretreatment with RAPA (Figure 5G).

## 3. Discussion

Many foods and cosmetics employ CIN as a fragrance because of its strong aroma. However, CIN can cause severe skin allergies and induce ACD, making it one of the 26 cosmetic fragrance allergens that are specified by EU regulations [28]. To develop treatment and prevention strategies for ACD, a deep investigation into the role of CIN in skin sensitization is necessary. The human cell line activation test (h-CLAT) adopted by the OECD can be used to evaluate the degree of DC activation by measuring the typical biomarkers of DC activation in THP-1 cells to estimate the skin sensitization potential under the framework of the AOP [29]. In this study, the allergic potential of CIN was evaluated via the h-CLAT method. We found that CIN exposure stimulated the activation of THP-1 cells, as indicated by increased CD54 and CD86 expression (Figure 1), and provoked oxidative stress in the cells. According to a previous study, which is consistent with our findings, CIN induced positive h-CLAT results upon detecting the expression of CD54 and CD86 [30,31]. These findings indicated that we successfully established a CIN-induced DC activation model. The connection between the autophagy process and the allergy-elicited DC response was identified via transcriptomics and proteomics [20]. Motivated by these findings, we conducted a transcriptome analysis and demonstrated that the *SQSTM1/p62*, *LC3B* and *ATG5* genes were differentially expressed following CIN stimulation in THP-1 cells and that the pathway with the greatest influence was autophagy (Figure 2). The verification experiments revealed that CIN increased the transcriptional and translational levels of the autophagy markers LC3B, p62, and ATG5 in THP-1 cells (Figure 3). This finding was consistent with a previous study in which contact sensitizers upregulated the gene and protein expression of LC3B and p62 in DCs [20,21,32]. Since autophagy has been identified as a defence or protective process of the cell, it is thought that allergens trigger active autophagy in DCs during skin sensitization [33]. Our results, therefore, revealed that autophagy might be involved in the regulation of THP-1 cell activation after CIN exposure.

Cellular autophagy is a conservative method of self-degradation, which is the process of lysosomal degradation and the reuse of damaged organelles and macromolecules [34,35]. Moreover, autophagy is closely related to intrinsic immunity, and these two essential pathways interact with each other [36,37,38]. Because autophagy has not been extensively studied in allergen-activated DCs, we decided to study the potential influence of autophagy on this process. To further explore the role of autophagy in CIN-induced THP-1 cell activation, the autophagy inhibitor Baf-A1 and activator RAPA were used in the present study. These results revealed that CIN and the late autophagy inhibitor Baf-A1 acted simultaneously on THP-1 cells and that *LC3B*, *p62* and *ATG5* accumulated due to blocked autophagic flux (Figure 4A–D). Moreover, the levels of the cell surface molecules CD54 and CD86, as well as the ROS content, were significantly increased in the presence of Baf-A1 (Figure 4E–G), suggesting an increase in the activation level of THP-1 cells. Wu et al. [39] investigated the mechanism of bisphenol A (BPA) toxicity via the autophagy inhibitor 3-methyladenine (3-MA) in RAW264.7 mouse macrophage cells and reported that the inhibition of autophagy led to significant upregulation of cellular inflammatory factors and further augmented BPA-induced cytotoxicity and damage to cells. Hegdekar et al. [40] reported that after adding autophagy inhibitors to RAW 246.7 cells, the levels of the inflammation-associated proteins NOS2 and NLRP3 were elevated and accompanied by an increase in nitric oxide production. Wang et al. [41] reported that an autophagy inhibitor counteracted the ameliorating effect of rosmarinic acid on the inflammatory response of RAW 264.7 cells and upregulated the expression of reactive oxygen species (ROS) and the secretion of proinflammatory factors. These experimental results reveal that the inhibition of autophagy facilitates xenobiotic-induced mononuclear-phagocyte activation, which is in accordance with our findings that autophagy blockade enhances CIN-induced THP-1 cell activation.

The autophagy activator chosen for this study was RAPA, an mTOR inhibitor. Because mTOR inhibits the induction of the autophagic process from the early stage of autophagosome formation to the late stage of lysosomal degradation, a reduction in mTOR expression significantly promotes autophagy [42,43]. Our results revealed that the inhibition of mTOR promoted autophagy after CIN exposure (Figure 5D). The expression of LC3B was significantly upregulated along with that of ATG5, and that of p62 was significantly downregulated due to autophagic degradation, further suggesting that autophagy was activated. In addition, THP-1 activation was significantly decreased, as indicated by reductions in CD54, CD86, and ROS levels (Figure 5E–G). Autophagy induction has been previously reported to reduce the immune cell inflammatory response. In previous publications, Wu et al. [39] and Hegdekar et al. [40] used RAPA to alleviate the adverse effects of autophagy inhibition on macrophages. Dai et al. [43] reported that treating THP-1-derived macrophages with the autophagy agonist resveratrol-activated cellular autophagy, significantly inhibited the high glucose-induced increase in ROS, and attenuated the inflammatory response [44]. Hence, autophagy stimulation can effectively relieve THP-1 cell activation caused by CIN. A previous study demonstrated that metformin, an autophagy activator, enhanced autophagic flux to inhibit macrophage activation in a DNFB-induced ACD mouse model [45]. These findings revealed that targeting autophagy might offer promising strategies for treating ACD.

Although the current study provides valuable insights into the role of autophagy in ACD, it is limited to an in vitro model. Future studies should utilize ACD animal models to validate these findings in vivo and examine the therapeutic effects of autophagy activators. Moreover, this study focused on autophagy-related molecules, and further exploration of the regulatory signalling pathways involved in autophagy is needed to provide a more comprehensive understanding of the molecular mechanisms involved in ACD.

## 4. Materials and Methods

### 4.1. Cell Culture and Reagents

The THP-1 human acute monocytic leukemia cell line was sourced from the Cell Resource Center at IBMS, CAMS/PUMC (Beijing, China). The cells were cultured in RPMI 1640 medium, enriched with 10% fetal bovine serum (FBS) and 1% penicillin/streptomycin. The cultures were maintained at 37 °C in a humidified incubator with 5% CO_2_. All cell culture reagents and phosphate-buffered saline (PBS) were purchased from Invitrogen (Carlsbad, CA, USA). Cinnamaldehyde (CIN, >98% purity), 3-(4,5-dimethylthiazol-2-yl)-2,5-diphenyltetrazolium bromide (MTT), and dimethyl sulfoxide (DMSO) were obtained from Sigma-Aldrich (St. Louis, MO, USA). Antibodies targeting LC3B, p62, ATG5, β-actin, and mTOR were provided by Abmart (Shanghai, China). Bafilomycin A1 (Baf-A1) was purchased from Sigma-Aldrich (St. Louis, MO, USA). Rapamycin (RAPA) was purchased from Selleck Chemicals (Houston, TX, USA).

### 4.2. Cell Viability Assay

Cytotoxicity was tested using the MTT assay. THP-1 cells were put in 96-well plates at a density of 1.6 × 10^5^ cells per well and treated with CIN concentrations from 0.01‱ (corresponding to 7.56 μM) to 1.00‱ (corresponding to 0.756 mM) for 24 h. After treatment, 10 µL of MTT solution (5 mg/mL) was added to each well and incubated at 37 °C for 4 h. After incubation, 100 µL of lysis solution, made of 5% isobutanol, 10% SDS, and 12 mM HCl in water, was added to each well to dissolve the formazan crystals. Absorbance was measured at 570 nm using a Tecan Infinite M200 Pro multimode microplate reader [14].

### 4.3. Flow Cytometry

THP-1 cells were plated in 24-well plates at a density of 1 × 10^6^ cells per well and treated with CIN for 24 h at concentrations ranging from 1.2 × CV75 to 0.335 × CV75, corresponding to a series of 1.2-fold dilutions. After incubation, cells were harvested and washed with PBS. For staining, the cells were incubated on ice for 30 min in the dark with fluorescence-conjugated monoclonal antibodies specific for CD54 (Biolegend, San Diego, CA, USA) and CD86 (BD Biosciences, San Jose, CA, USA), along with an IgG1 isotype control antibody. Following incubation, the cells were washed three times with staining buffer (PBS supplemented with 2% FBS) before being analyzed by flow cytometry using an Accuri C6 system (BD Biosciences, San Jose, CA, USA). Nonviable cells, identified by low forward and side scatter, were excluded from analysis, and a minimum of 10,000 events per sample were recorded. The relative fluorescence intensity (RFI), serving as an indicator of CD54 and CD86 expression, was calculated in accordance with the OECD test guideline (No. 442E) [14].

### 4.4. Determination of Intracellular Reactive Oxygen Species (ROS)

Intracellular ROS generation was assessed using the 6-carboxy-2′,7′-dichlorofluorescein diacetate (DCFH-DA) fluorescence probe from the ROS assay kit (Beyotime, Shanghai, China). THP-1 cells were treated with CIN at concentrations of 0.69‱ (corresponding to 0.522 mM) and 1‱ (corresponding to 0.756 mM) for 2 h and then incubated with DCFH-DA solution at 37 °C for 20 min. After incubation, the cells were washed three times with a serum-free culture medium to remove any excess DCFH-DA that had not been internalized. Intracellular ROS levels were quantified by measuring fluorescence intensity using flow cytometry (BD Accuri C6, BD Biosciences).

### 4.5. Transcriptomics

THP-1 cells were plated in 24-well plates at a density of 2 × 10^6^ cells per well and cultured with different doses of CIN: 0.69‱ (corresponding to 0.522 mM) and 1‱ (corresponding to 0.756 mM) for 6 h. After incubation, the cells were transferred into 1.5 mL non-enzymatic centrifuge tubes and centrifuged at 8000 rcf for 2 min at 4 °C. The supernatant was discarded, and the pellet was resuspended. Then, 1 mL of TransZol Up was added to each tube, thoroughly mixed, and stored at −80 °C. Then, RNA was extracted for analysis. The entire transcriptome analysis was conducted by Novo to Source Biotechnology (Beijing, China). Briefly, the Illumina platform was used for sequencing data filtering and assembly, while edgeR was employed to identify differentially expressed genes (DEGs). Volcano plots of DEGs and functional enrichment analysis were performed using Magic Novogene, with Gene Ontology (GO) database annotations. Each treatment was performed in triplicate with three independent biological replicates.

### 4.6. Real-Time PCR Measurement

Trizol reagent (Invitrogen, Carlsbad, CA, USA) was used to extract total RNA from THP-1 cells. As directed by the manufacturer, one microgram of RNA was transformed into complementary DNA (cDNA) using the ReverTra Ace qPCR RT Kit (TOYOBO, Osaka, Japan)and a Mastercycler^®^ EPgradient (Eppendorf, Hamburg, Germany). RT-PCR was performed using a SYBR Green Real-time PCR Master Mix (TOYOBO, Osaka, Japan), and the LightCycler^®^ 480 II system sourced from Roche (Basel, Switzerland). After 30 s of initial denaturation at 95 °C, 40 cycles of 95 °C for 5 s, 55 °C for 10 s, and 72 °C for 15 s comprised the amplification conditions. The Ct values served as the basis for real-time quantification, and the 2^−ΔΔCt^ method was used to calculate gene expression levels that were adjusted to β-actin.

### 4.7. Western Blot Analysis

THP-1 cells were treated with CIN at 0.69‱ (corresponding to 0.522 mM) and at 1‱ (corresponding to 0.756 mM) for 24 h. Cells were lysed with cell lysis buffer for western and immunoprecipitation (IP) (Beyotime, Shanghai, China), and cell debris was removed via centrifugation. The concentrations of protein from the supernatant were quantified using the BCA Protein Assay Kit (Beyotime, Shanghai, China). After quantitation, equal amounts of protein were loaded on a sodium dodecyl sulphate–polyacrylamide gel (SDS–PAGE; Beyotime, Shanghai, China) and transferred into a polyvinylidene fluoride (PVDF) membrane (Beyotime, Shanghai, China). The membrane was blocked with a QuickBlock™ Western Occluder (Beyotime, Shanghai, China) and incubated at 1:1000 primary antibody overnight at 4 °C. The membranes were treated with a horseradish peroxidase-conjugated (HRP) secondary antibody (Santa Cruz, CA, USA) for 2 h at a diluted concentration of 1:5000 at room temperature and were visualized using a Tanon-5200 Multi System (Tanon, Shanghai, China). The expression level of each protein was analyzed using ImageJ software version 1.54g (National Institutes of Health, Bethesda, MD, USA) and normalized to β-actin.

### 4.8. Statistical Analysis

Statistical analyses were performed using GraphPad Prism software version 8.0.2 (Dotmatics, Boston, MA, USA). For comparing multiple CIN treatment doses with the control group, a one-way ANOVA followed by Dunnett’s post hoc test was used. The interactions between Baf-A1 and CIN, as well as RAPA and CIN, on THP-1 cells, were assessed using a two-way ANOVA with Bonferroni post hoc correction. All experiments were conducted in triplicate, and data are presented as the mean ± SD from three independent trials. Statistical significance was considered for *p*-values less than 0.05.

## 5. Conclusions

Autophagy plays a crucial role in CIN-induced activation of THP-1 cells. CIN treatment stimulated the expression of CD54 and CD86 and enhanced intracellular oxidative stress in THP-1 cells, indicating an activated dendritic cell (DC) phenotype. Additionally, the upregulation of the expression of LC3B, p62, and ATG5, which are key autophagy markers, further confirmed the involvement of autophagy in this process. Autophagy was shown to modulate both activation and redox homeostasis in CIN-treated THP-1 cells, as illustrated in Figure 6. This study provides new insights into the autophagy pathway, highlighting its importance in regulating CIN-induced DC activation. Given its potential role in the pathogenesis of ACD, targeting autophagy in DCs may offer a novel approach to the prevention and treatment of this disease. Future research focusing on the regulatory mechanisms of autophagy and DC activation is essential for the development of more targeted therapeutic strategies for ACD.

## Figures and Tables

**Figure 1 ijms-26-02377-f001:**
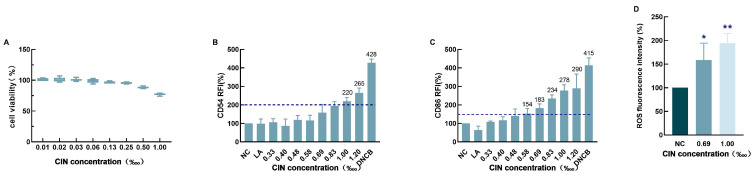
THP-1 cells experience oxidative stress and cell activation brought on by CIN. (**A**) The MTT test was used to assess the cytotoxic effect of CIN on THP-1 cells following a 24 h treatment period. THP-1 cells were exposed to CIN for 24 h in order to measure cell activation markers. Using flow cytometry, the expression levels of CD54 (**B**) and CD86 (**C**) were measured. The results are shown as relative fluorescence intensity (RFI) in relation to the control group, and the number above the bar is the mean RFI of the indicated group which meets the positive threshold. The dashed lines indicate the positive cut-off values for CD54 (200) and CD 86 (150), respectively. After two hours of CIN treatment, THP-1 cells’ intracellular ROS level (**D**) was determined by flow cytometry. The mean ± SD of three independent experiments is represented in the data. When compared to the control group, * *p* < 0.05 and ** *p* < 0.01.

**Figure 2 ijms-26-02377-f002:**
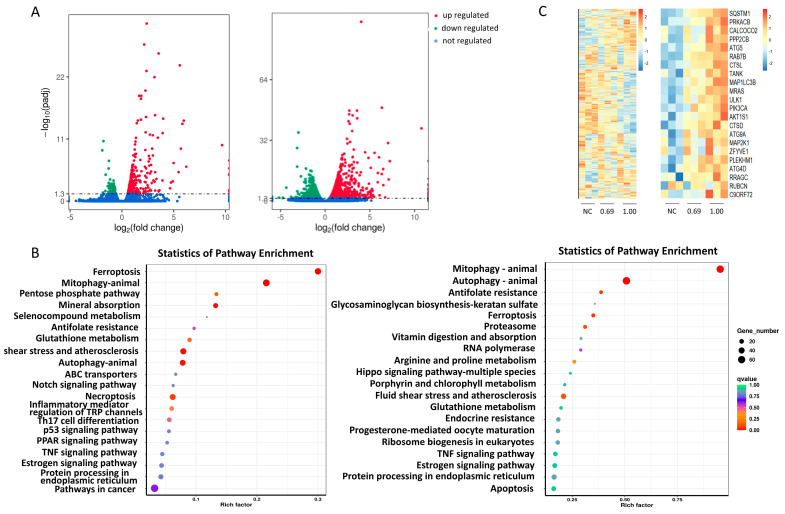
Transcriptome analysis of THP-1 cells after CIN treatment. (**A**) The number of differentially expressed genes identified in the transcriptomes of THP-1 cells treated with 0.69‱ CIN (**left**) and 1‱ CIN (**right**). The dashed lines represent the significance level. (**B**) A scatterplot showing the KEGG pathway enrichment of differentially expressed genes in the transcriptomes of THP-1 cells treated with 0.69‱ CIN (**left**) and 1‱ CIN (**right**). (**C**) Differentially expressed genes from the transcriptome sequencing results (**left**) and the expression of autophagy-associated genes (**right**) were analyzed.

**Figure 3 ijms-26-02377-f003:**
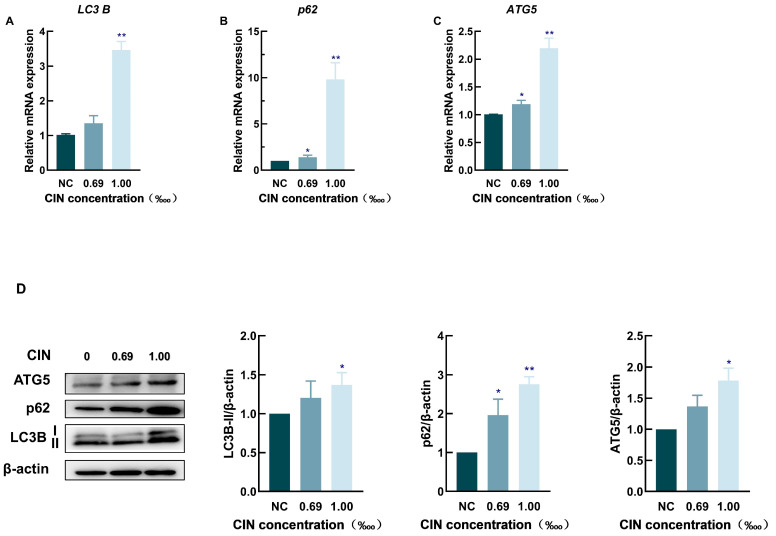
Effect of CIN on the expression of autophagy-related pathway genes and proteins. The mRNA levels of *LC3B* (**A**), *p62* (**B**), and *ATG5* (**C**) were measured in CIN-treated THP-1 cells. (**D**) Protein expression levels of LC3B, p62, and ATG5 were analyzed. Protein expression changes were quantified relative to the untreated control (set at 1) and normalized to β-actin. Data are representative of three independent experiments, with results presented as mean ± SD. * *p* < 0.05, ** *p* < 0.01, compared with the control group.

**Figure 4 ijms-26-02377-f004:**
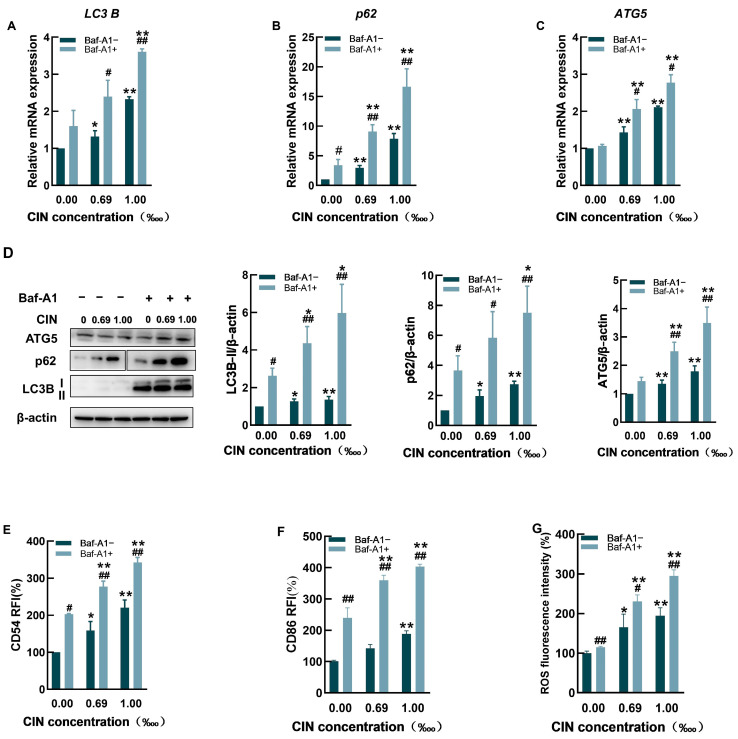
Effects of Baf-A1 on CIN-treated THP-1 cells. The mRNA levels of *LC3B* (**A**), *p62* (**B**), and *ATG5* (**C**) were measured in CIN-treated THP-1 cells with (+) or without (−) Baf-A1. (**D**) Protein expression of LC3B, p62, and ATG5 was assessed in CIN-treated THP-1 cells pretreated with Baf-A1. Protein expression changes were quantified relative to the untreated control (set at 1) and normalized to β-actin. The expression of CD54 (**E**), CD86 (**F**), and ROS (**G**) were determined in CIN-treated THP-1 cells with (+) or without (−) Baf-A1. Data are representative of three independent experiments, with results presented as mean ± SD. * *p* < 0.05, ** *p* < 0.01 compared with the control group; # *p* < 0.05, ## *p* < 0.01 compared with the −Baf-A1 group with the same treatment.

**Figure 5 ijms-26-02377-f005:**
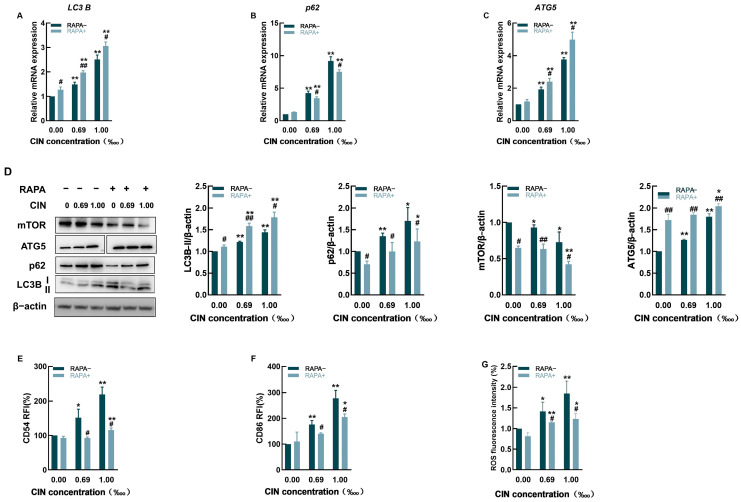
Effects of RAPA on CIN-treated THP-1 cells. The mRNA levels of *LC3B* (**A**), *p62* (**B**), and *ATG5* (**C**) were measured in CIN-treated THP-1 cells with (+) or without (−) RAPA. (**D**) Protein expression levels of LC3B, p62, mTOR, and ATG5 were assessed in CIN-treated THP-1 cells pretreated with RAPA. Protein expression changes were quantified relative to the untreated control (set at 1) and normalized to β-actin. The expression of CD54 (**E**), CD86 (**F**), and ROS (**G**) was determined in CIN-treated THP-1 cells with (+) or without (−) RAPA. Data are representative of three independent experiments, with results presented as mean ± SD. * *p* < 0.05, ** *p* < 0.01 compared with the control group; # *p* < 0.05, ## *p* < 0.01 compared with the −RAPA group with the same treatment.

**Figure 6 ijms-26-02377-f006:**
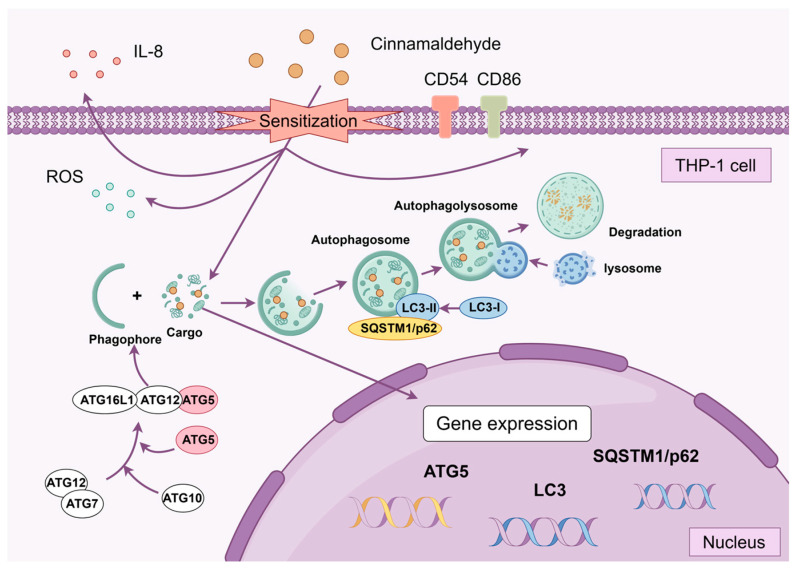
Schematic representation of the effects of autophagy on CIN-induced THP-1 cell activation.

## Data Availability

The data that support the findings of this study are available on request from the corresponding author.

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
