# Peer review of "Modulation of Autophagy on Cinnamaldehyde Induced THP-1 Cell Activation"

_ijms, 2025, doi:10.3390/ijms26062377_

Round 1
Reviewer 1 Report
Comments and Suggestions for Authors
My comments:
1. There are unexplained abbreviations in the text (e.g. THP-1) - review the manuscript from this angle again and fill in the missing explanations of the abbreviations used
2. in-text citation: missing space before square brackets - insert them
3. Figure 2B is illegible, fonts are too small - improve the graphic quality of the figure
4. The introduction is a very general introduction to the issue, it is not possible to determine the reason for undertaking this research based on it - supplement the introduction with a more detailed description; explain the autophagy phenomenon
5. line 299 "pro-tein"- typographical error - correct; search the entire manuscript for typographical errors and correct them
6. indicate the weaknesses and limitations of your study (experiment)
7. format the reference list according to MDPI guidelines
8. reference [8] - provide more detailed parameters for this citation
9. When you write about ROS, which specific ROS do you mean? - explain
Author Response
请参阅附件

Reviewer 2 Report
Comments and Suggestions for Authors
Major Comments
· The study is well-conducted and provides insights into the role of autophagy in CIN-induced activation of dendritic cell-like THP-1 cells. However, the novelty needs to be clearly articulated in comparison to previous studies on autophagy in allergic contact dermatitis (ACD). Highlighting the unique findings in this work would strengthen its impact.
· While the transcriptomic and proteomic analyses provide a comprehensive dataset, the authors should elaborate more on the downstream signaling pathways linking autophagy markers (LC3B, p62, and ATG5) to the observed phenotypic changes in THP-1 cells. Does CIN modulate specific transcription factors or kinases in this context?
· The study is limited to in vitro THP-1 models. Considering the clinical relevance of ACD, a discussion on how these findings could translate to in vivo systems would add depth. Are there plans to validate these findings using animal models.
· The CV75 for CIN was calculated, but the rationale for choosing specific concentrations (e.g., 0.69% and 1%) for subsequent experiments should be further clarified, particularly concerning their physiological relevance.
· While the use of Baf-A1 and rapamycin is appropriate, the dual effects of these compounds on autophagy and other cellular pathways could confound results. A control experiment using alternative autophagy modulators would strengthen conclusions.
Minor Comments
· The abstract is clear but could benefit from including quantitative findings (e.g., specific changes in CD54 and CD86 expression levels).
· Replace "autophage" with "autophagy" for consistency.
· In the introduction, provide more background on the clinical implications of cinnamaldehyde as a fragrance allergen. Citing real-world data on ACD cases caused by CIN would be beneficial.
· Explain the relevance of THP-1 cells as a model for dendritic cell studies in greater detail for readers unfamiliar with this context.
· Ensure all figures are labeled clearly. For instance, the y-axis in Figure 1D should explicitly state the units of ROS fluorescence intensity.
· In Figure 2, the scatter plot and heatmap for KEGG pathway enrichment could benefit from larger font sizes for readability.
· In the Discussion, Include a brief discussion on potential therapeutic strategies targeting autophagy, such as the use of rapamycin analogs in ACD management.
· The role of oxidative stress in the context of autophagy modulation could be elaborated further.
· Some references, such as those for OECD guidelines and previous studies on autophagy, are relevant but dated. Include more recent citations to reflect advancements in the field.
· Replace “authophage” with “autophagy” throughout the manuscript.
· Ensure consistent use of formatting for gene and protein names (e.g., LC3B should be italicized for the gene but not the protein).
Reviewer 3 Report
Comments and Suggestions for Authors
The manuscript "Modulation of autophagy on cinnamaldehyde induced THP-1 cell activation" by ME et al., is interesting pointing out that cinnamaldehyde can affect THP-1 cell activation. The results are interesting, however some experimental set-ups are missing form the manuscript and have to be implemented in the revised manuscript.
Please in the introduction section elaborate on the components LC3B, p62, ATG5 to be more suitable for readers to follow the context.
Line 78, dramatically is to strong expression for cell survival of about 75%? Please refine the sentence.
Line 295, Please change "For full day" with "For 24 hours".
Line 279, Please explain how you selected the range of CIN concentrations for experiments?
Please add molar concentrations of CIN which were applied to cells.
Line 279, Please provide details on centrifugation as well as content of the Lysis buffer.
Line 300, It is not clear on which portions authors mean?
Lines 302-309 is a repetition of previous paragraph.
In the experimental part there are no details in what solution CIN was solubilized, and which amounts were added in the cell culture.
Data presented in Figure 2 is not visible for readers.
Details on Western blot analysis is missing, as the experiment can not be repeated by readers.
Comments on the Quality of English LanguageCould be slightly improved.
Round 2
Reviewer 1 Report
Comments and Suggestions for Authors
Dear authors - "THP-1" is the name of a cell line, but at the same time it is an abbreviation - "Tohoku Hospital Pediatrics-1". This name (like any other) came from something - please include it in the text.
In addition, photos of the original blots are illegible. These photos are also not original, they are just pasted into Word. Please attach real original photos that have not been edited in any way, including changing the format.
Besides, your answer is generally a bit rude. Moreover - I found something like "请参阅附件" in your answer and unfortunately it is not understandable to me. I recommend avoiding using a language other than English in this type notes.
Reviewer 3 Report
Comments and Suggestions for Authors
The manuscript has been substantially improved and should be accepted for publication after minor correction.
Please translate a part (lines 316-319) in methodology how the experiments been performed, not how they should be done.
Round 3
Reviewer 1 Report
Comments and Suggestions for Authors
I have no other comments. Thank you.